# Electromyography as an Objective Outcome Measure for the Therapeutic Effect of Biofeedback Training to Reduce Post-Paralytic Facial Synkinesis

**DOI:** 10.3390/healthcare13050550

**Published:** 2025-03-04

**Authors:** Isabell Hahnemann, Julia Fron, Jonas Ballmaier, Orlando Guntinas-Lichius, Gerd Fabian Volk

**Affiliations:** 1Department of Otorhinolaryngology, Jena University Hospital, Am Klinikum 1, 07747 Jena, Germany; isabell.hahnemann@freenet.de (I.H.); julia.fron@med.uni-jena.de (J.F.); jonas.ballmaier@med.uni-jena.de (J.B.); orlando.guntinas@med.uni-jena.de (O.G.-L.); 2Facial-Nerve-Center Jena, Jena University Hospital, Am Klinikum 1, 07747 Jena, Germany; 3Center of Rare Diseases Jena, Jena University Hospital, Am Klinikum 1, 07747 Jena, Germany

**Keywords:** facial palsy, biofeedback, electromyography, synkinesis, rehabilitation, post-paralytic facial synkinesis, outcome measure, patient-reported outcome measure

## Abstract

Biofeedback rehabilitation for facial palsy is not yet routinely available for patients. **Methods**: To improve evidence, the effect of an intensive two-week facial training combining electromyography (EMG) and visual biofeedback training of 30 patients (76.7% female; median age: 48.6 years) with post-paralytic facial synkinesis was objectively evaluated. At the beginning of each training day, EMG amplitudes of both halves of the face were recorded during relaxation using the EMG system that was synchronously used for the EMG biofeedback training. A single-factor analysis of variance was performed for the change over time, and a *t*-test was used to evaluate the side differences. **Results**: At the end of the training program, there was a significant decrease in the EMG amplitudes of both halves of the face (synkinetic side: *p* < 0.001; contralateral side *p* = 0.003), indicating an improved voluntary muscle relaxation. There was also a significant improvement in Sunnybrook Facial Grading System, Facial Disability Index and Facial Clinimetric Evaluation scores, which were assessed before the start of training and at the end (*p* < 0.001). **Conclusion**: Electrophysiological improvements can be objectively measured using surface EMG.

## 1. Introduction

In around 20–30% of patients with Bell’s palsy or in any case of more severe facial nerve lesion, facial palsy does not heal completely. Instead, an aberrant reinnervation process develops, which is usually characterized by the occurrence of synkinesis [1,2,3,4]. This is an unintentional hyperactivation and co-movement of the facial muscles that occurs simultaneously with intentional facial movements [2]. For example, movements of the corner of the mouth can lead to ipsilateral eye closure in the sense of oro-ocular synkinesis. Ultimately, any facial muscle in any combination with other facial muscles can be affected [4].

The development of synkinesis requires an axonal damage. During regeneration, not all of the regrowing axons reach their original muscle to be innervated, but many sprout undirected into other peripheral branches of the facial nerve. Synkinesis occurs when an axon or several axons with the same function send sprouts to muscle fibers from two or more different muscles with different physiological functions, i.e., to the original target muscle and to any other muscle. This excessive muscle innervation can also lead to an increased activity and hypertonic state [2,5]. The physical disabilities are often accompanied by psychological impairments due to stigmatization and non-verbal communication [2,6,7].

In addition to drug treatment with botulinum toxin, facial training is also considered as a treatment option for synkinesis [8]. However, there are currently no international standard protocols for facial training or rehabilitation [2]. One possible training principle is biofeedback, which has been used as a combination of visual and electromyography (EMG) feedback at Jena University Hospital since 2012 to treat patients with post-paralytic facial synkinesis [9]. The EMG biofeedback signals provide information about the state of activity of the facial muscles. Under the guidance of a specialized therapist, patients learn to perform symmetrical facial movements according to the interpretation of the biofeedback signals and to voluntarily avoid synkinesis occurring at the same time [2]. In the last years, we were often surprised that many patients with chronic facial palsy with synkinesis were not aware about the increased muscle activity of their affected side of the face. In addition, their physicians and therapists were focusing their treatment on strengthening of “paretic” muscles, similar to the treatment for stroke patients. To demonstrate to these patients that their affected muscles are not too weak but instead often over-active and a result of aberrant reinnervation, EMG biofeedback is often a very convincing technical approach. In contrast to visual feedback (mirror, video), EMG offers the first objectifiable measurement unit in the form of electronically measurable (instead of purely visually assessable) amplitudes.

The therapeutic principle of biofeedback has been used clinically for more than 50 years in the rehabilitation of neuromuscular disorders. Reviews have shown a positive therapeutic effect specifically for the use of biofeedback in rehabilitation after strokes, fecal and urinary incontinence, and in the treatment of headaches [10]. For facial nerve palsy, nevertheless, the evidence for the use of biofeedback in reviews is unclear due to the poor comparability of the individual studies. The use of a combination of EMG and visual feedback also makes it difficult to attribute therapy effects to the individual biofeedback modality in the evaluation [10,11,12]. This also explains why, in a recently published Delphi method-based study, the facial therapy experts surveyed attached little importance to the use of biofeedback training in the rehabilitation of facial nerve palsies and did not rate it as an essential therapy option [13].

This is all the more astonishing because the therapy effect of the intensive 10-day facial training was already demonstrated in previous studies as an improvement in the facial nerve palsy-related quality of life, as subjectively assessed by the patients [14], as well as by the grading objectively assessed by a trained examiner using the Sunnybrook Facial Grading System (SFGS) [9]. Starting from this, in the present study, the training effect should be visualized by the change in the amplitudes of the routinely applied surface EMG biofeedback system as an objective outcome measure. It is hypothesized that relaxation of the facial muscles reflected by a reduction in the EMG amplitudes is a positive effect of this training.

The aim is not only to record externally visible changes that occur after completing the combined biofeedback training, but also to measure direct electrophysiological changes in the facial muscles.

## 2. Materials and Methods

### 2.1. Study Design, Ethical Approval, Eligibility Criteria and Study Protocol

This prospective observational and longitudinal study included patients with post-paralytic facial synkinesis of different etiologies who participated in the intensive 2-week facial training combining electromyography and visual biofeedback conducted at the Facial-Nerve-Center of the Department of Otorhinolaryngology, Jena University Hospital, between April and July 2022. This study was conducted according to the guidelines of the Declaration of Helsinki of 1975 and approved by the Ethics Committee of the Jena University Jena, Germany (protocol code 2022-2589_1-BO; approved on 5 April 2022). All participants gave written informed consent prior to their inclusion in this study. There was no difference in treatment of the patients whether participating in the study or not.

The inclusion criteria were as follows: age ≥ 18 years; a unilateral, chronic, peripheral facial palsy with stable symptoms for more than 6 months [1] with needle electromyography (EMG)-confirmed voluntary activity in the affected facial muscles including aberrant, synkinetic reinnervation; participation in a two-week biofeedback-based training of facial muscles at the Facial-Nerve-Center, Jena University Hospital; and the provision of written consent.

The exclusion criteria were as follows: acute facial nerve palsy; lack of signs of synkinetic reinnervation of the facial musculature in needle EMG; botulinum toxin treatments within the last 3 months before the start of training; missing participation, discontinuation or interruption in EMG-biofeedback training for more than 2 training-days at the Facial-Nerve-Center Jena; and lack of written consent from the patient or lack of capacity to consent [6,14].

The 2-week biofeedback training is based on elements of biofeedback combined with Taub’s Constraint Induced Movement Therapy (CIMT) [6]. The aim of this facial muscle training is to improve facial symmetry and muscle coordination, reduce motor deficits, and enable relaxation in the face to avoid muscles interacting and limiting each other’s movements. By these physical improvements, psychosocial impairments are reduced as well. Usually, two patients were trained simultaneously on two consecutive weeks from Monday to Friday for three hours in the morning under the guidance of one of the two specialized therapists (hereinafter referred to as therapist A and B) at the Facial-Nerve-Center Jena, using a 2-channel surface EMG and visual biofeedback. For this purpose, two bipolar 24 × 30 mm foam hydrogel electrodes of the type Kendall H124SG (Cardinal Health, Dublin and Odio) were attached to each side of the face, recording the muscles of the cheek and mouth region. These electrodes were coupled with the NeXus-10 MKII biofeedback system (Mind Media, Roermond-Herten, The Netherlands). One of the bipolar electrodes was placed over the arcus zygomaticus halfway between the angulus oculi lateralis and the ventral attachment of the auricle, the second in the area of the modiolus anguli oris on the cheek (Figure 1). If the electrodes needed to be repositioned from areas with pronounced scars, wrinkles or a beard, they were placed as close as possible to the original position and then symmetrically on the contralateral side of the face to avoid visual confusion for patients. Using BioTrace+ software (version V2018) animations (Mind Media BV, The Netherlands), the EMG activity levels of both halves of the face, which are proportional to the muscle activity, were then visualized numerically and as vertical bars on a screen in front of the patient. Furthermore, a video-generated mirror image of the patient’s face was generated via a webcam and displayed in the middle of the screen. The therapist sitting opposite could also see the EMG feedback bars and mirror images of the patients on their screen (Figure 1) [9].

Patients should learn to perform facial movements more symmetrically while consciously avoiding too strong movements of the contralateral side of the face and controlling unintended co-movements of other ipsilateral facial muscles in the sense of synkinesis. The aim is to equalize the activity level of both halves of the face during the exercises and to improve the conscious relaxation phases of the facial muscles between the individual exercises. Similar to Taub’s principles, simpler exercises are initially learned in the first week of training with continuous feedback from the therapist. These are then combined into more complex movement sequences in the second week (Table 1).

Patients were also instructed to correct movements independently using the biofeedback signals [15]. Patients were also given homework to repeat previously trained exercises alone without their therapist in the afternoons and at the weekend for about two hours with a hand mirror. Their training units and performance were documented by the patients themselves in a training diary. This should prepare the patients for regular self-motivated training after the instructed two weeks of training. A follow-up examination took place six months after the intensive training to evaluate the long-term effects and the success of the therapy [9,14,15].

### 2.2. Measurement Times and Recorded Parameters

All grading assessments were performed at two time points: T1 = one day before the start of the training (mostly on a Sunday before the training started on Monday), and T2 = on the afternoon of the day before the training ended (mostly on Thursday afternoon of the second training week) after completion of the second-last training day (Figure 2). Of the 8–10 days on which the patients trained on site with biofeedback support, 7–9 training days between T1 and T2 were examined as part of the study.

At the first examination date (T1), all patients were personally informed about the study and patient history data were recorded. Furthermore, the facial palsy-related quality of life was assessed by the patients’ self-assessment using the Facial Clinimetric Evaluation Scale (FaCE) and the Facial Disability Index (FDI) in the German version [15]. With the help of the Sunnybrook Facial Grading System [16], an additional assessment of facial nerve palsy was carried out by a non-blinded independent rater using sets of 12 photographs of standardized facial movements taken at each date (T1 and T2). These photographs were taken in a highly standardized way in a photo studio by an independent professional photographer as part of the clinical routine. Finally, the patients were instructed to document the amplitudes of the surface EMG at a muscle relaxation task routinely performed daily during the beginning of their training on forms specially designed for this study.

The training was individually adapted for each patient according to the personal therapy goals, motivation, and severity of the synkinesis and facial motion impairments of the paresis. Due to the expected influence of the individualization on the training effects, an additional training documentation was carried out for the study. At the beginning of each training day, the patients routinely performed a muscle relaxation task and documented three EMG amplitudes in µV for each side of the face (Figure 3).

To avoid mistakes in the self-documentation, all patients received written instructions (Figure A1). They were contacted by telephone on the first days of training and reinstructed if necessary. The first author was also in close contact with the therapists, who were also informed about the documentation forms and were, therefore, able to support the patients in completing them. To avoid fatigue effects, the amplitudes were recorded at the beginning of the training. When evaluating the documentation forms of each patient, the three EMG amplitudes recorded for each side of the face were transferred directly to the electronic data collection files of the study.

On the first (T1) and second examination dates (T2), the patients completed the FaCE and FDI questionnaires. The completed training documentation forms were collected, and the standardized photo series (containing 12 photos each) were taken to assess the severity of facial nerve palsy using the Sunnybrook Facial Grading System.

### 2.3. Facial Clinemetric Evaluation and Facial Disability Index

The disease-specific Facial Clinimetric Evaluation Scale (FaCE) and the Facial Disability Index (FDI) in the validated German-language version were used to record the facial nerve palsy-related quality of life. In both questionnaires, the patient self-assesses the limitations in quality of life resulting from the paresis [16].

The scores are calculated according to the formulas given on the form of the German-language questionnaire, with the exception of the FaCE Facial Comfort Score. Low scores indicate a high level of restrictions on quality of life [17,18].

### 2.4. Sunnybrook Facial Grading System

With the help of the Sunnybrook Facial Grading System (FSGS), an additional assessment of facial nerve palsy was carried out by a non-blinded, but independent rater (I. H.) observing a series of twelve standardized photographs taken routinely and independently of the study by a professional photographer in a photo studio before the start of training and after completion of the penultimate training day. To minimize the risk of a bias, the rating of the sets of facial movement photographs were not carried out by the therapists or physicians involved in the clinical treatment of the patients, but by an independent medical student (I. H.). The following nine facial expressions were photographed: (1) at rest, (2) frowning, (3) closing both eyes gently, (4) closing both eyes with maximum force, (5) wrinkling the nose, (6) raising both corners of the mouth with the mouth closed, (7) showing teeth, (8) pursing the lips, (9) blowing cheeks, (10) snarling, (11) pulling down both corners of the mouth, and (12) smiling naturally [9]. The validated German version of this grading instrument was also used in this case [19]. The scores were calculated using the formulas given on the assessment form, with 100 points corresponding to normal facial function and 0 points corresponding to complete facial nerve palsy [20]. Alternative grading systems such as the House–Brackmann Score or the Stennert Parese Index, which also serve to assess the functionality and symmetry of the face, were not used in the international setting due to the recommendation to use the Sunnybrook Facial Nerve Grading System as the standard [21].

### 2.5. Statistics

Statistical analysis was performed using SPSS version 25.0 (IBM, Armonk, NY, USA). Our case number analysis, based on the effect sizes of the previous studies, resulted in a patient number of 30. Descriptive statistics were used for continuous data (mean, standard deviation, median, minimum and maximum) and qualitative data (frequencies) [22]. EMG amplitudes of the two halves of the face were evaluated daily. Due to scheduling issues, measurements were grouped by training days (1–9) instead of specific weekdays for better comparability. To assess the side difference (healthy vs. diseased side of the face) in EMG amplitudes, a paired *t*-test [23] was used, assuming normal distribution [24,25,26]. Effect size was measured using Cohen’s d (small: >0.2, medium >0.5, large: >0.8) [27]. To examine the training effects over time, ANOVA with repeated measures tested significant changes in EMG amplitudes across the training days. To analyze the individual influence of therapists, participants were grouped into three categories: (1) therapist A throughout; (2) therapist B throughout; (3) change of therapist within the training period. The correlation between EMG amplitudes changes and therapist categories was examined using ANOVA with repeated measures, considering therapist influence as a covariate. The effect size for the therapist influence was measured using eta-square (medium: _ > 0.3, strong: _ > 0.5 [26]. All tests were two-sided with significance set at *p* < 0.05.

## 3. Results

### 3.1. Patients’ Characteristics

A total of 30 patients were included (76.7% female; median age at start of the training: 48.62 ± 12.41 years, Table 2). A variety of etiologies of facial nerve palsy were recorded in the patient population. With 12 cases (40.0%), idiopathic palsy occurred most frequently, followed by 9 cases (30.0%) occurring after removal of a benign tumor. On average, patients were able to start the intensive biofeedback training 2.80 ± 2.41 years after the initial diagnosis (range: 1.12 to 10.99 years). Twelve patients (40.0%) completed only eight training days between T1 and T2, and three patients (10.0%) completed only seven training days of the maximum nine possible observed training days between the first and second measurement dates due to public holidays and organizational delays on the day of admission.

### 3.2. Facial Grading and Facial Palsy-Related Quality of Life

With regard to the facial palsy-related quality of life of the patients, the facial paresis training resulted in significantly higher FDI physical and FDI social function scores and a higher FaCE total score, with a medium effect size. Furthermore, the training also resulted in a significant increase in the mean value of the SFGS score, with a large effect size (Figure 4 and Table A1).

### 3.3. EMG Amplitudes

Figure 5 shows reductions in the mean values of the EMG amplitudes of both halves of the face between the first and the last training day for all 30 patients. This change was significant for both the synkinetic (*p* < 0.001; d = 0.92) and contralateral (*p* = 0.003; d = 0.59) side, with a much higher effect size of Cohen’s d on the synkinetic side (0.92 > 0.59). The EMG amplitudes of the synkinetic side were significantly higher than those of the contralateral side at both time points (before and after training). This side difference, which was higher on the synkinetic side at the beginning of training (*p* < 0.001; d = 1.07), was lower but still present on the last day of training (*p* < 0.001; d = 0.78).

As shown in Figure 6, the significant lateral difference in the EMG amplitudes of both halves of the face, which existed on the last training day of the first week before the weekend (*p* < 0.001; d = 0.72), was also detectable after the weekend on the first training day of the second week (*p* < 0.001; d = 0.59), with at least a medium effect size. Furthermore, there was a trend towards larger mean values of EMG amplitude for both halves of the face after the mid-training weekend.

When comparing the EMG amplitudes of both halves of the face in pairs on each training day (Figure 7), it is noticeable that up to and including the eighth day of training, there was a significant difference between the synkinetic and contralateral side for the patient cohort under consideration, with at least medium to large effect sizes for Cohen’s d. The EMG amplitudes of the synkinetic side were significantly higher than those of the contralateral side. After completion of the ninth day of training, no significant side difference (*p* = 0.058) could be determined, with a medium effect size (Cohen’s d = 0.53). However, at this late time point in the training, only n = 15 patients were still participating in the study, so the lack of significance is mostly explained by the lower number of subjects.

On the synkinetic side, there was a significant reduction in the EMG amplitudes of the pre-training measurements on the first training day compared to all other days, which indicates the training progress. No significant change could be demonstrated for any other possible combination of training days two to nine (Table 3).

On the contralateral side, significant reductions in EMG amplitudes could be measured from the first to the third, fifth and seventh to ninth day of training, which indicates the training progress. Further significant differences could be demonstrated for the following combinations of training days: days 2 and 5; days 2 and 8, days 5 and 6, and days 6 and 9 (Table 4).

As shown in Table 5, no significant influence could be demonstrated for either the instruction therapist or a change in therapist during the training on the EMG amplitudes of the synkinetic (*p* = 0.891) and contralateral (*p* = 0.450) side of the face.

## 4. Discussion

Similar to previous studies, a significant improvement in facial nerve palsy-related quality of life was also demonstrated for the patient cohort of the present study after completing the biofeedback training using the established patient-reported outcome measures FaCE and FDI [14]. Again, a significant improvement in the SFGS score, as assessed by the objective grading by an examiner independent of the training, was also demonstrated [9]. Compared to other subjective measurement instruments, such as the House–Brackmann Scale, the SFGS is currently considered the most suitable because it can also be used to record very small changes in synkinesis after a therapeutic intervention. Another advantage is the very good intra- and inter-rater reliability [21]. Dalla Toffola et al. also used the SFGS and were able to demonstrate that both isolated training with a mirror and isolated training with EMG biofeedback are effective. There was no significant difference between the two therapy groups with or without mirror [28].

Based on this, the results of this study provide further insights into the pathophysiology of the therapeutic effect of combined biofeedback training in facial nerve palsy. Over the course of the nine observed training days, there was a significant decrease in the muscular tension of both halves of the face, as quantified via surface EMG during relaxation. The mean values of the EMG amplitudes of the affected side were significantly greater than those of the contralateral side. Only at the end of the training, after 9 days, when the training group was reduced to n = 15 patients, no significant side difference (*p* = 0.058) could be determined. In this respect, the biofeedback training at the Facial-Nerve-Center in Jena University Hospital should not be shortened, with currently 10 training days supervised by specialized therapists.

At the start of the second week of training after the weekend, an increase in amplitudes was recognizable on both sides of the face, although this did not prove to be significant. Apparently, a break from training with the EMG feedback system over the weekend does not impair the training progress significantly, but it does at least reverse the trend of reduced EMG values from training day to training day. Furthermore, the choice of therapist or a change of therapist during the training course has no influence on the change in EMG amplitudes over the training period. This can be interpreted as a sign of the good standardization and reproducibility of our training program.

In other studies, a further possibility for objectifying the training effects of rehabilitation programs for facial palsy was demonstrated by measuring changes in the opening width of the eyes on the basis of still images from video recordings of facial movements. For mirror-supported training, a reduction in asymmetries in the intervention group compared to the control group without therapy during synkinetic movements could be demonstrated [29,30]. The authors are aware that, by examining EMG amplitudes during relaxation in this study, they have not contributed to a more uniform view of the therapeutic effects of facial paresis biofeedback training.

Also, in contrast to other studies, this study had no control group with another therapeutic interventions, such as physiotherapy, or without any therapeutic intervention due to the complex implementation of the partial inpatient facial biofeedback training [31,32,33]. As this biofeedback training is based on a fixed combination of visual and EMG-supported biofeedback, it was not possible to attribute the proven therapeutic effects to a single biofeedback modality. Furthermore, no comparative estimate of the training effect to that of other forms of therapy is possible. In future research, randomized trials comparing biofeedback training with other therapies (e.g., physiotherapy) or no treatment, respectively, using specialized training methods with either visual or EMG-based biofeedback training, are needed.

In addition, it must be pointed out that the examined patient group has very heterogeneous characteristics (e.g., etiology, time since diagnosis). Analyses of subgroups with matching patient characteristics are recommended in the future in order to be able to assess the therapy effect even more specifically.

In line with the results of the review by Franz et al. [34], it was shown that the amplitudes of the surface EMG in an open derivation are easy to record in everyday therapy and represent a further possibility for objectively recording therapy effects over time. The low invasiveness of this measurement method and the lower dependence of the results on the examiner compared to grading are also advantageous. In addition, the method offers the possibility of not only recognizing externally visible changes, but also recording the electrophysiological changes that occur directly in the facial musculature after completing biofeedback training for facial palsy. This shows that the recording of EMG amplitudes can also be used to record training successes within short observation periods. In the long term, the patients have to integrate their exercises into everyday life on their own, and a further EMG-recording of the mimic muscles at the follow-up appointment after six months would be interesting for the investigation of long-term therapy effects in further studies.

## 5. Conclusions

It was shown that electrophysiological changes in the facial musculature and thus also the therapeutic effects of combined biofeedback training in patients with post-paralytic synkinetic facial nerve palsy can be objectively recorded by means of an openly visible surface EMG recording.

In future, patients could record the changes in EMG amplitudes during relaxation, as in the study, in order to record changes that occur directly during training and thus better understand the success of treatment.

This finding may be useful for future randomized controlled studies. By extending the two-channel EMG to a multi-channel examination, electrophysiological changes could be specifically examined for individual facial muscles in the future. The use of imaging techniques such as sonography or MRI, but also 3D videos or quantitatively analyzed photo series, would also be conceivable for the objective detection of structural changes in the facial musculature following biofeedback training.

## Figures and Tables

**Figure 1 healthcare-13-00550-f001:**
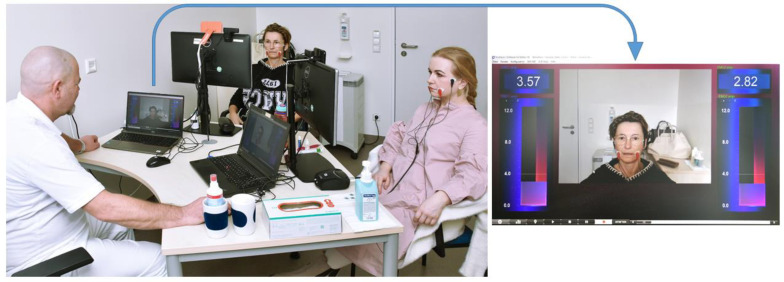
Training setting. The typical training setup with two patients and a therapist sitting opposite is shown on the left side. The activity of the facial muscles is recorded using electromyography via adhesive electrodes and displayed on a screen to the left and right of a webcam-based mirror image as a two vertical feedback bars and numbers. Patients and therapist see the same screens.

**Figure 2 healthcare-13-00550-f002:**
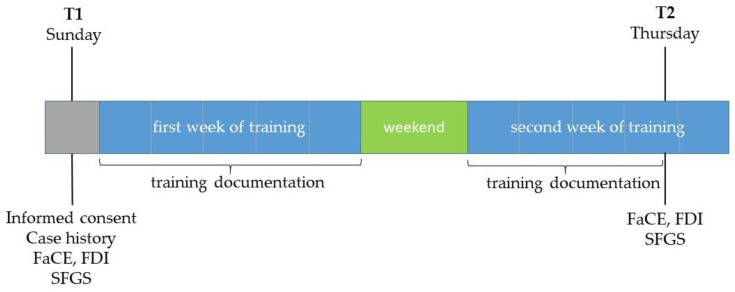
Overview of measurement times and recorded parameters (T1: first examination date; T2: second examination date; FaCE: Facial Clinemetric Evaluation; FDI: Facial Disability Index; SFGS: Sunnybrook Facial Grading System). Grey: assessment day; blue: intervention period; green: no intervention.

**Figure 3 healthcare-13-00550-f003:**
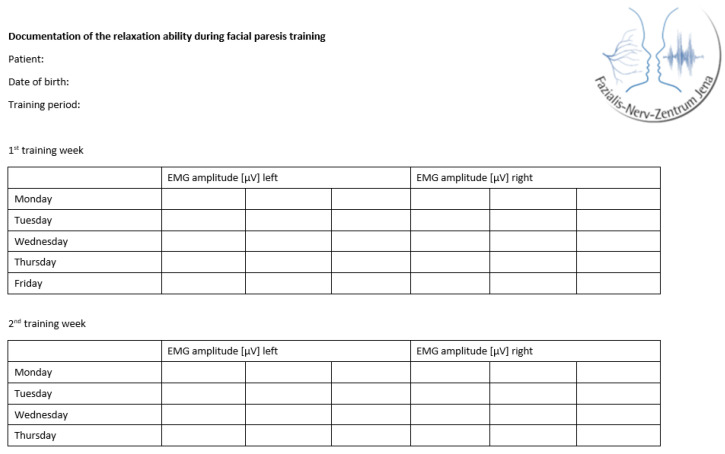
Tabular template to document the ability to relax.

**Figure 4 healthcare-13-00550-f004:**
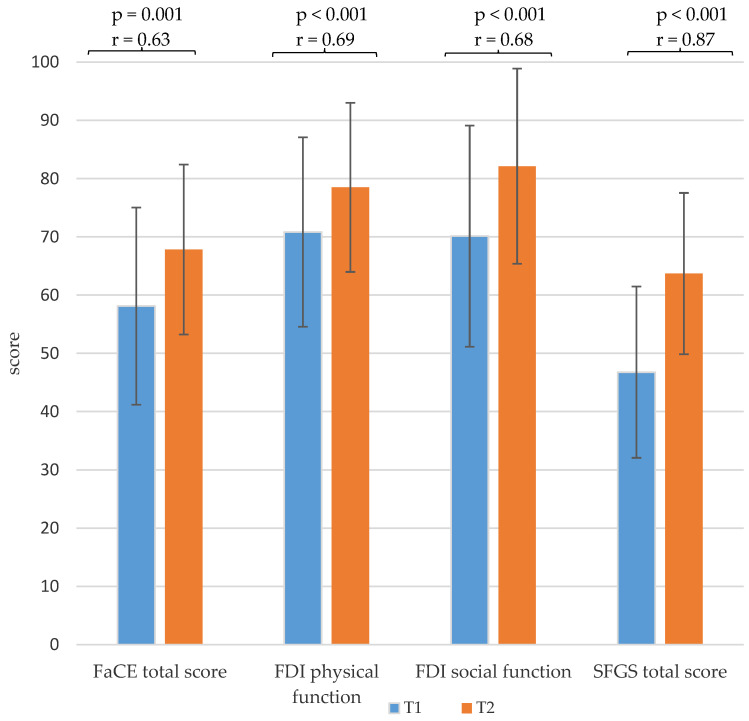
Comparison of the mean values and standard deviation of the sum scores of the Facial Clinimetric Evaluation (FaCE), the Facial Disability Index (FDI) and the Sunnybrook Facial Grading System (SFGS) of the baseline measurement before the start of the training (T1) and the second measurement at the end of the training (T2) with *p*-values and effect size r.

**Figure 5 healthcare-13-00550-f005:**
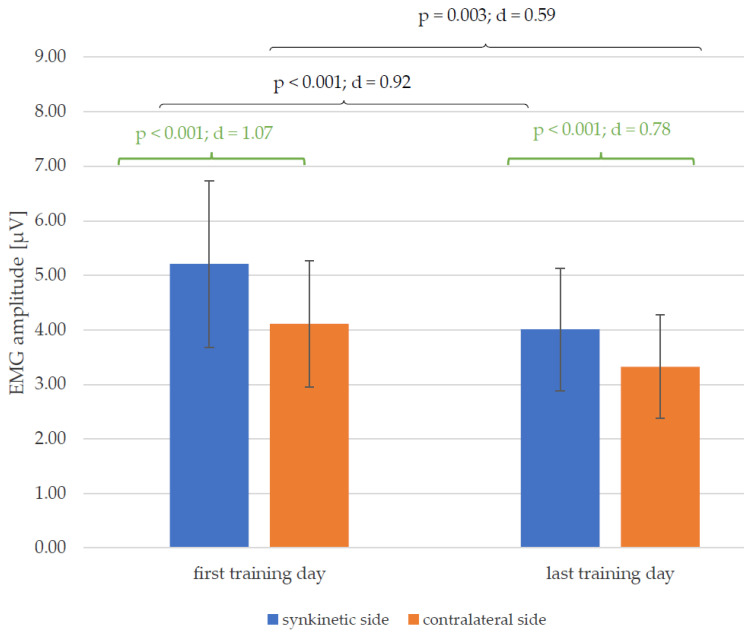
Changes in the mean values and standard deviation of the EMG amplitudes from the first to the last training day (black) and the side differences (green) on the first and last training day, with *p*-values and Cohen’s d for all 30 patients.

**Figure 6 healthcare-13-00550-f006:**
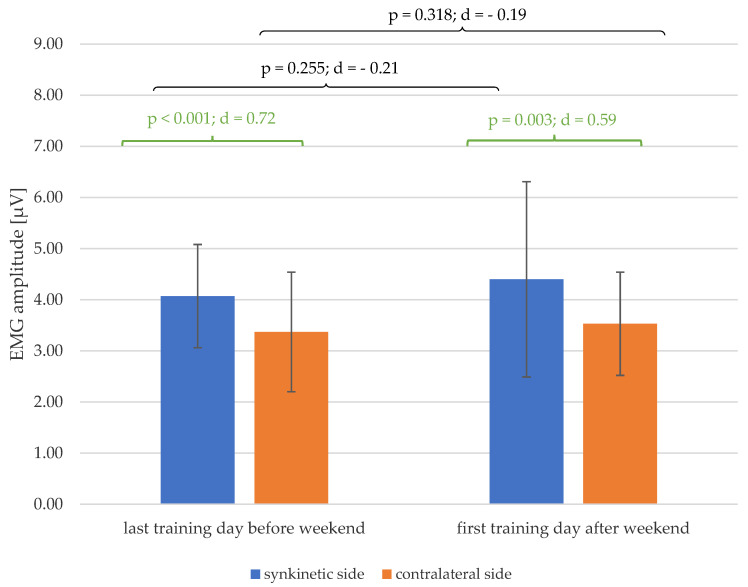
Changes in the mean values and standard deviation of the EMG amplitudes over the mid-training weekend (black) and the side differences (green) on the days before and after the weekend, with *p*-values and Cohen’s d.

**Figure 7 healthcare-13-00550-f007:**
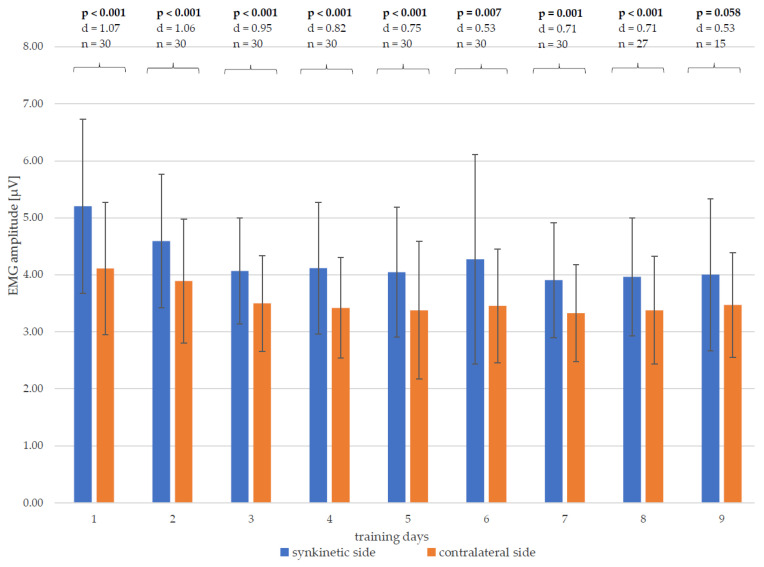
Changes in the mean values and standard deviation of the EMG amplitudes from each training day (days 1–9) of both halves of the face and the side differences, with *p*-values and Cohen’s d; significant results marked in bold; n—number of patients.

**Table 1 healthcare-13-00550-t001:** Overview of the most frequently performed facial paresis training exercises.

List of Exercises
Long eyelid closure
Closed smile
Open smile
Pursed lips/kissing mouth
Closed smile with transition to kissing mouth
Pursed lips with transition to showing teeth
Open smile with transition to “O”/”U”
Letter exercise O/U
Letter exercise E/I
Letter exercise A
Alternating one-sided smile, then both sides
Show teeth
Wrinkle nose
Raise/lower eyebrows
Puff out cheeks

**Table 2 healthcare-13-00550-t002:** Patients’ characteristics.

Parameter	Absolute (N)	Relative (%)
All	30	100.0
Gender		
Male	7	23.3
Female	23	76.7
Handedness		
Right	30	100.0
Left	0	0
Etiology		
idiopathic	12	40.0
postoperative with benign tumor	9	30.0
Ramsay Hunt syndrome	6	20.0
traumatic	3	10.0
congenital	0	0
postoperative with malignant tumor	0	0
cerebral insult	0	0
Localization of the paresis		
right	15	50.0
left	15	50.0
Presence of oro-ocular synkinesis		
yes	30	100.0
no	0	0
Therapist week 1		
Therapist A	10	33.3
Therapist B	20	66.7
Therapist week 2		
Therapist A	19	63.3
Therapist B	11	36.7
Number of training days completed between the first (T1) and second (T2) measurement point		
9 training days	15	50.0
8 training days	12	40.0
7 training days	3	10.0
	Mean ± SD	Median, range
Age at beginning of facial palsy in years	45.88 ± 13.71	47.33; 18.45–70.31
Age at start of training in years	48.62 ± 12.41	48.84; 24.05–71.43
Height in m	1.71 ± 0.09	1.70; 1.56–1.89
Weight in kg	78.90 ± 17.84	72.5; 48.0–150.0
BMI in kg/m^2^	26.08 ± 6.43	24.95; 16.61–43.36
Period between beginning of facial palsy and reinnervation in years (n = 28 *)	0.33 ± 0.25	0.27; 0.04–1.03
Time between beginning of facial palsy and first contact with Facial-Nerve-Center in years	2.11 ± 2.30	1.21; 0.44–9.79
Time between beginning of facial palsy and first measurement (T1) in years	2.80 ± 2.41	1.84; 1.12–10.99

* Period unknown for n = 2 patients.

**Table 3 healthcare-13-00550-t003:** *p*-values (top row, significant results marked in bold) of the changes in EMG amplitudes of the synkinetic side for each possible combination of training days, with 95% confidence intervals in brackets.

	1	2	3	4	5	6	7	8	9
1		**0.017** (0.2–1.7)	**0.001** (0.6–1.9)	**0.018** (0.2–2.1)	**0.002** (0.6–2.2)	**0.011** (0.3–1.9)	**0.003** (0.6–2.4)	**0.001** (0.7–2.2)	**0.001** (0.7–2.1)
2	**0.017** (0.2–1.7)		0.202 (−0.8–0.2)	0.549 (−1.0–0.6)	0.118 (−1.0–0.1)	0.570 (−0.8–0.5)	0.121 (−1.3–0.2)	0.066 (−1.1–0.0)	0.097 (−1.0–0.1)
3	**0.001** (0.6–1.9)	0.202 (−0.2–0.8)		0.764 (−0.5–0.7)	0.483 (−0.5–0.2)	0.536 (−0.3–0.6)	0.382 (−0.8–0.3)	0.172 (−0.5–0.1)	0.564 (−0.7–0.4)
4	**0.018** (0.2–2.1)	0.549 (−0.6–1.0)	0.764 (−0.7–0.5)		0.374 (−0.7–0.3)	0.879 (−0.7–0.8)	0.327 (−1.0–0.3)	0.305 (−0.9–0.3)	0.516 (−1.0–0.5)
5	**0.002** (0.6–2.2)	0.118 (−0.1–1.0)	0.483 (−0.2–0.5)	0.374 (−0.3–0.7)		0.340 (−0.3–0.8)	0.688 (−0.7–0.5)	0.601 (−0.4–0.3)	0.938 (−0.6–0.6)
—	**0.011** (0.3–1.9)	0.570 (−0.5–0.8)	0.536 (−0.6–0.3)	0.879 (−0.8–0.7)	0.340 (−0.8–0.3)		0.052 (−0.7–0.0)	0.117 (−0.8–0.1)	0.232 (−0.8–0.2)
7	**0.003** (0.6–2.4)	0.121 (−0.2–1.3)	0.382 (−0.3–0.8)	0.327 (−0.3–1.0)	0.688 (−0.5–0.7)	0.052 (0.0–0.7)		0.900 (−0.4–0.5)	0.669 (−0.3–0.5)
8	**0.001** (0.7–2.2)	0.066 (0.0–1.1)	0.172 (−0.1–0.5)	0.305 (−0.3–0.9)	0.601 (−0.3–0.4)	0.117 (−0.1–0.8)	0.900 (−0.5–0.4)		0.795 (−0.4–0.6)
9	**0.001** (0.7–2.1)	0.097 (−0.1–1.0)	0.564 (−0.4–0.7)	0.516 (−0.5–1.0)	0.938 (−0.6–0.6)	0.232 (−0.2–0.8)	0.669 (−0.5–0.3)	0.795 (−0.6–0.4)	

**Table 4 healthcare-13-00550-t004:** *p*-values (top row, significant results marked in bold) for the changes in EMG amplitudes of the contralateral side for each possible combination of training days, with 95% confidence intervals in brackets.

	1	2	3	4	5	6	7	8	9
1		0.075 (−0.1–1.0)	**0.005** (0.2–1.0)	0.050 (0.0–1.2)	**0.006** (0.3–1.6)	0.128 (−0.2–1.2)	**0.018** (0.2–1.4)	**0.009** (0.2–1.3)	**0.006** (0.3–1.4)
2	0.075 (−0.1–1.0)		0.608 (−0.6–0.4)	0.652 (−0.8–0.5)	**0.031** (−0.9–0.0)	0.916 (−0.6–0.5)	0.198 (−0.9–0.2)	0.084 (−0.7–0.0)	0.059 (−0.8–0.0)
3	**0.005** (0.2–1.0)	0.608 (−0.4–0.6)		0.935 (−0.4–0.4)	0.147 (−0.8–0.1)	0.714 (−0.4–0.6)	0.409 (−0.7–0.3)	0.377 (−0.6–0.2)	0.158 (−0.7–0.1)
4	0.050 (0.0–1.2)	0.652 (−0.5–0.8)	0.935 (−0.4–0.4)		0.076 (−0.7–0.0)	0.694 (−0.5–0.7)	0.460 (−0.7–0.3)	0.507 (−0.7–0.4)	0.210 (−0.7–0.2)
5	**0.006** (0.3–1.6)	**0.031** (0.0–0.9)	0.147 (−0.1–0.8)	0.076 (0.0–0.7)		**0.039** (0.0–0.8)	0.467 (−0.2–0.5)	0.262 (−0.1–0.5)	0.584 (−0.2–0.3)
6	0.128 (−0.2–1.2)	0.916 (−0.5–0.6)	0.714 (−0.6–0.4)	0.694 (−0.7–0.5)	**0.039** (−0.8–0.0)		0.052 (−0.6–0.0)	0.153 (−0.7–0.1)	**0.033** (−0.7–0.0)
7	**0.018** (0.2–1.4)	0.198 (−0.2–0.9)	0.409 (−0.3–0.7)	0.460 (−0.3–0.7)	0.467 (−0.5–0.2)	0.052 (0.0–0.6)		0.851 (−0.3–0.3)	0.672 (−0.4–0.3)
8	**0.009** (0.2–1.3)	**0.084** (0.0–0.7)	0.377 (−0.2–0.6)	0.507 (−0.4–0.7)	0.262 (−0.5–0.1)	0.153 (−0.1–0.7)	0.851 (−0.3–0.3)		0.441 (−0.4–0.2)
9	**0.006** (0.3–1.4)	0.059 (0.0–0.8)	0.158 (−0.1–0.7)	0.210 (−0.2–0.7)	0.584 (−0.3–0.2)	**0.033** (0.0–0.7)	0.672 (−0.3–0.4)	0.441 (−0.2–0.4)	

**Table 5 healthcare-13-00550-t005:** Interaction between the change in EMG amplitude and the therapists.

	*p*-Value	Partial η²
Interaction between the change in EMG amplitude of the synkinetic side and therapist A, therapist B and change of therapists	0.891	0.365
Interaction between the change in EMG amplitude of the contralateral side and therapist A, therapist B and change of therapists	0.450	0.603

Partial η²—partial eta square.

## Data Availability

The raw data supporting the conclusions of this article will be made available by the authors, without undue reservation.

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
