# Peer review of "Electromyography as an Objective Outcome Measure for the Therapeutic Effect of Biofeedback Training to Reduce Post-Paralytic Facial Synkinesis"

_healthcare, 2025, doi:10.3390/healthcare13050550_

Round 1
Reviewer 1 Report
Comments and Suggestions for Authors
1. title seems to be too long and complicated to understand. any chance to make it more simple?
2. line 54, "jena". is a city, university or hospital? hard to understand without authors' affiliations (which most of the readers don't think to read first hand)
3. line 184, " (Error! Reference source not found.)"
4. eq 1: what are those items? are they from app a?(which is not clearly understand). give their exact names. also why only items 4, 6, and 13 are used? the other items on App A seems to be useless in this approach? why did you divide to 4 at the formula? it seems that you have to explain Eq 1 better and more briefly
5. is the sample size statistically adequate?
6. apart for variance analysis, aren't there any more statistical tests can be applied to approve your approach?
Author Response
1. title seems to be too long and complicated to understand. any chance to make it more simple:
-> Thank you for highlighting this important point. We changed the title to:
EMG as Objective Outcome Measure for the Therapeutic Effect of Biofeedback Training to reduce Post-Paralytic Facial Synkinesis
Do you prefer it that way?
2. line 54, "jena". is a city, university or hospital? hard to understand without authors' affiliations (which most of the readers don't think to read first hand)
-> Very important point. We changed in the whole document "Jena" to "Jena University Hospital".
3. line 184, " (Error! Reference source not found.)"
-> In line 184, no publication war referenced but only our Figure 3.
4. eq 1: what are those items? are they from app a?(which is not clearly understand). give their exact names. also why only items 4, 6, and 13 are used? the other items on App A seems to be useless in this approach? why did you divide to 4 at the formula? it seems that you have to explain Eq 1 better and more
-> Sorry for not explaining it in a clear way. We both questionaris, FaCE and FDI, like discriped in the original publications:
https://onlinelibrary.wiley.com/doi/epdf/10.1097/00005537-200103000-00005
https://academic.oup.com/ptj/article/76/12/1288/2632957?login=true
To make it as easy as possible, both original papers are attached as pdf-portfolio. Please see the attachment.
5. is the sample size statistically adequate?
-> Our case number analysis, based on the effect sizes of the previous studies, resulted in a patient number of 30. We used the effect size from studies using evaluations based on photos and PROMs. Having found significant results even with this small sample size emphasizes the effect size of our intervention. We also included the case number analysis in our text.
6. apart for variance analysis, aren't there any more statistical tests can be applied to approve your approach?
-> After consultation with our department for Medical Statistics, the test we used are the best choice to work with our type of study. But to clarify our results, we improved the description of our statistical methods and our results. We hope to have solves your concerns on that way.

Reviewer 2 Report
Comments and Suggestions for Authors
The manuscript contributes to the field; however, I recommend the following improvements to enhance its quality, readability, and impact:
Title:
The authors are asked to simplify and clarify the title by removing repetition and highlighting the study's principal features.
Abstract:
The authors are requested to remove redundant information and focus on emphasizing the most critical information.
Introduction:
The authors are requested to highlight the medical significance of synkinesis and its effects on both physical and psychological health. Furthermore, they must elucidate the justification for employing EMG-based biofeedback as a treatment technique.
Materials and Methods:
To improve readability, authors should combine ethical approval, inclusion criteria, and training protocol into a single subsection. They should additionally simplify statistical test descriptions by highlighting their purpose such as analyzing training effects, therapist influence, and EMG amplitude while being clear and concise.
Results:
The statistics report is very detailed, obscuring the primary findings. The authors are asked to simply refine this part to emphasize the most significant results while providing an organized review of important outcomes, keeping clarity and emphasis in the narrative.
Discussion:
The authors are encouraged to expand on the study's limitations, particularly the lack of a control group and the difficulty in differentiating between the effects of visual biofeedback and those based on EMG. They might suggest overcoming these constraints in future research via randomized trials or specialized training methods. The authors are asked to emphasize their contribution to the field and its prospective influence on clinic
Conclusions:
The authors should explain the importance of the noticed electrophysiological changes and the therapeutic advantages of biofeedback training.
Author Response
Title:
The authors are asked to simplify and clarify the title by removing repetition and highlighting the study's principal features.
-> Thank you for highlighting this important point. We changed the title to:
EMG as Objective Outcome Measure for the Therapeutic Effect of Biofeedback Training to reduce Post- Paralytic Facial Synkinesis
Do you prefer it that way?
Abstract:
The authors are requested to remove redundant information and focus on emphasizing the most critical information.
-> We removed redundant information and focused more on the essential information.
Introduction:
The authors are requested to highlight the medical significance of synkinesis and its effects on both physical and psychological health. Furthermore, they must elucidate the justification for employing EMG-based biofeedback as a treatment technique
-> That is a very important point and we have improved your Introduction to highlight it more.
Why do we use EMG instead of easier available techniques? In the last years, we were often surprised, that many patient with chronic facial palsy with synkinesis were not aware about the increased muscle activity of their affected side of the face. In addition, their physicians and therapists were focusing their treatment in strengthening of “paretic” muscles similar to the treatment for stroke patients. To demonstrate these patients, that their affected muscles are not too weak but instead often over-active as and result of aberrant reinnervation, EMG biofeedback is often a very convincing technical approach. Of cause, similar results can be achieved by palpation of the muscles or detailed visual inspections of faces.
Materials and Methods:
To improve readability, authors should combine ethical approval, inclusion criteria, and training protocol into a single subsection. They should additionally simplify statistical test descriptions by highlighting their purpose such as analyzing training effects, therapist influence, and EMG amplitude while being clear and concise.
-> Thanks for helping us to improve this weakness of our manuscript. We have re-written it in that way:
Statistical analysis was performed using SPSS version 25.0. Descriptive statistics were used for continuous data (mean, standard deviation, median, min, max) and qualitative data (frequencies). EMG amplitudes of the face were evaluated daily, but due to scheduling issues, measurements were grouped by training days (1-9) instead of specific weekdays for better comparability.
To assess the side difference in EMG amplitudes (diseased vs. healthy side), a paired t-test was used, assuming normal distribution. Effect size was measured using Cohen's d (small: ≥0.2, medium: ≥0.5, large: ≥0.8). To examine training effects over time, ANOVA with repeated measures tested significant changes in EMG amplitudes across training days.
To analyze the influence of therapists, participants were grouped into three categories: 1) therapist A throughout, 2) therapist B throughout, and 3) therapist change mid-training. The correlation between EMG amplitude changes and therapist categories was examined using ANOVA with repeated measures, considering therapist influence as a covariate. Effect size for therapist influence was measured using eta-square (medium: ≥0.3, strong: ≥0.5). All tests were two-sided, with significance set at p < 0.05.
Results:
The statistics report is very detailed, obscuring the primary findings. The authors are asked to simply refine this part to emphasize the most significant results while providing an organized review of important outcomes, keeping clarity and emphasis in the narrative.
-> Due to the novelty of using EMG not only for training but also outcome measure to quantify the therapeutic effect, we would like to keep your description of the results so broad and detailed. Anyway, we can add a summery for the clinicians: Our analysis have shown that from day to day the differences between affected and non-affected side of the face get smaller but don´t disappear completely at the end of our training.
Due to the novelty of using EMG not only for training but also outcome measure to quantify the therapeutic effect, we would like to keep your description of the results so broad and detailed. Anyway, we can add a summery for the clinicians: Our analysis have shown that from day to day the differences between affected and non-affected side of the face get smaller but don´t disappear completely at the end of our training.
Discussion:
The authors are encouraged to expand on the study's limitations, particularly the lack of a control group and the difficulty in differentiating between the effects of visual biofeedback and those based on EMG. They might suggest overcoming these constraints in future research via randomized trials or specialized training methods. The authors are asked to emphasize their contribution to the field and its prospective influence on clinic
->Thanks for this valuable comment. We included the points your mentioned in the improved text.
Furthermore, no comparative estimate of the training effect to that of other forms of therapy is possible. In future research randomized trials comparing biofeedback training with other therapies (e.g., physiotherapy) or no treatment respectively specialized training methods with either visual or EMG-based biofeedback training are needed. In addition, it must be pointed out that the examined patient group has very heterogeneous characteristics (e.g. etiology, time since diagnosis). Analyses of subgroups with matching patient characteristics are recommended in the future in order to be able to assess the therapy effect even more specifically.
Conclusions:
The authors should explain the importance of the noticed electrophysiological changes and the therapeutic advantages of biofeedback training.
->Thanks for this valuable comment. We included the points your mentioned in the improved text.
In future, patients could record the changes in EMG amplitudes during relaxation, as in the study, in order to record changes that occur directly during training and thus better understand the success of treatment.
Reviewer 3 Report
Comments and Suggestions for Authors
The manuscript evaluates the effects of a two-week electromyography (EMG)-based biofeedback training program on patients with post-paralytic facial synkinesis. The authors claimed EMG-based biofeedback training to significantly reduces synkinesis and improves facial function and quality of life. However, a few points need to be justified to better support the current conclusions.
1. The major concern was the lack of a control group in the study when comparing biofeedback training with other therapies (e.g., physiotherapy) or no treatment? Also, how does the absence of a randomized controlled trial impact the interpretation of results?
2. Was there potential bias due to the non-blinded evaluation of the Sunnybrook Facial Grading System? Can the author describe if there are any additional outcome measures (e.g., patient satisfaction or long-term adherence) strengthen the study?
3. The sample size of 30 patients might not be sufficient to draw robust conclusions given the heterogeneity in etiology and time since diagnosis. An increased sample size with well-aligned population characteristics is needed to support the current findings and the generalizability of the conclusions in broader populations with different severities or causes of facial palsy?
Author Response
1. The major concern was the lack of a control group in the study when comparing biofeedback training with other therapies (e.g., physiotherapy) or no treatment? Also, how does the absence of a randomized controlled trial impact the interpretation of results?
->Thanks for this valuable comment. We included the points your mentioned in the improved text.
Furthermore, no comparative estimate of the training effect to that of other forms of therapy is possible. In future research randomized trials comparing biofeedback training with other therapies (e.g., physiotherapy) or no treatment respectively specialized training methods with either visual or EMG-based biofeedback training are needed. In addition, it must be pointed out that the examined patient group has very heterogeneous characteristics (e.g. etiology, time since diagnosis). Analyses of subgroups with matching patient characteristics are recommended in the future in order to be able to assess the therapy effect even more specifically.
2. Was there potential bias due to the non-blinded evaluation of the Sunnybrook Facial Grading System? Can the author describe if there are any additional outcome measures (or long-term adherence) strengthen the study?
-> To reliability of the Sunnybrook Facial Grading System is high, so that it can be used as a clinical outcome measure. Anyway, if we would use the Sunnybrook Facial Grading System as our primary outcome measure, we would also prefere blinded raters. In a previous study from us using photos to quantify the therapeutic effect instead of EMG, the rater was blinded. Here, we are happy to use the objective EMG measurement to show that not only photos, PROMs and our own impressions are positive, but also objective outcome measures.
3. The sample size of 30 patients might not be sufficient to draw robust conclusions given the heterogeneity in etiology and time since diagnosis. An increased sample size with well-aligned population characteristics is needed to support the current findings and the generalizability of the conclusions in broader populations with different severities or causes of facial palsy?
-> Yes, like very often, a bigger sample size would be useful to analysis sub-groups and questions that are more specific. Nevertheless, if you take the heterogeneity of our patients in etiology and time since onset of facial palsy in account, the strong effects we have found are even more impressive and show how strong the clinical relevance of such a specialised facial training is. Another study with much more patients, but focussing only on PROMs as outcome measures, is under preparation. The statistical analysis show very similar effects like in this study. Also in previous photo-based evaluations of our training concept, we have seen comparable effects (Volk, G.F.; Roediger, B.; Geißler, K.; Kuttenreich, A.M.; Klingner, C.M.; Dobel, C.; Guntinas-Lichius, O. Effect of an Intensified Combined Electromyography and Visual Feedback Training on Facial Grading in Patients With Post-paralytic Facial Synkinesis. Front Rehabil Sci 2021, 2, 746188, doi:10.3389/fresc.2021.746188.).
Round 2
Reviewer 1 Report
Comments and Suggestions for Authors.
Author Response
We are very happy for your positive feedback, Reviewer 1.
Thank you for speaking out so clearly in favor of publishing our manuscript.
Reviewer 2 Report
Comments and Suggestions for Authors
This paper has been significantly improved and is now acceptable.
Author Response
We are very happy for your positive feedback, Reviewer 2.
Thank you for speaking out so clearly in favor of publishing our manuscript.